# Preparation and Characterization of Porous Carbon Composites from Oil-Containing Sludge by a Pyrolysis-Activation Process

**Wen-Tien Tsai *** and **Yu-Quan Lin**

Graduate Institute of Bioresources, National Pingtung University of Science and Technology,
Pingtung 912, Taiwan; wsx55222525@gmail.com
* Correspondence: wttsai@mail.npust.edu.tw; Tel.: +886-8-7703202

**Abstract:** Large amounts of oil-containing sludge (OS) are produced in the energy, industry and services sectors. It is mainly composed of residual oil and particulate matter, thus posing an environmental threat and leading to resource depletion if it is improperly handled. In this work, the OS feedstock was thermally treated to produce porously magnetic carbon composites (CC) using a pyrolysis-activation process. Using the data on the thermogravimetric analysis (TGA) of the OS feedstock, the thermal activation experiments were performed at 850 °C as a function of residence time (30, 60 and 90 min). The results of pore analysis for the resulting CC products showed that the Brunauer–Emmett–Teller (BET) surface area greatly decreased from 21.59 to 0.56 $m^2/g$ with increasing residence time from 30 to 90 min. This decline could be associated with continuous gasification by $CO_2$, thus causing the removal of limited carbon on the surface of CC for a longer reaction time. Furthermore, the physical properties of the resulting CC products can be enhanced by post acid-washing due to the development of new pores from the leaching-out of inorganic minerals. The BET surface area increased from 21.59 to 40.53 $m^2/g$ at the residence time of 30 min. Obviously, the resulting CC products were porous materials with mesopores and macropores that were concurrently formed from the thermal activation treatment. These porous features were also observed by scanning electron microscopy (SEM).

**Keywords:** oil-containing sludge; thermal activation; Fe/C composite; acid-washing; pore analysis

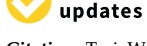



## 1. Introduction

Specialty oils are used to work machines and vehicles in the process and transportation industries. During maintenance and repair, the related equipment regularly generates oil-containing waste or deteriorated oils, which are further managed for storage, transport and uses such as lubricating oil [1,2] and engine oil [3,4]. Another source of oil-containing waste comes from the tank bottom sludge in petroleum refineries, which also contains large components of heavy hydrocarbons oils, mud sand, inorganic salts, and mechanical impurities (e.g., iron-containing rusts) [5]. Because of the residual oils that remain, the collected waste is valorized to convert it into available resources such as auxiliary fuel oils. The so-called oil-containing sludge is produced from the oil-sludge separation processes in the off-site oil-containing waste or deteriorated oils treatment plant.

Obviously, the oil-containing sludge still contains heavy residual oils and inorganic matters such as sand and rusts. With consideration for resource recycling and the circular economy, the valorization methods of oil-containing sludge have been investigated by using resource processing technologies [6], including solvent extraction, pyrolysis, emulsification, freezing/thawing, ultrasonication, flotation, and centrifugation. On the other hand, few reports have focused on the production of carbon materials from oily sludge by activation processes. However, the samples in these studies were mostly original sources from drilling stations and petroleum refinery plants, and not derived from recycling plants. In addition, these studies focused on the activation process in the production of porous

carbon materials from the oil-containing sludge by chemical agents, including, potassium hydroxide [7–10], zinc chloride [11], phosphoric acid [12], sodium oxide [13], and ferric chloride [11]. However, a preparation method involving chemical activation is not environmentally and economically promising one because it produces considerable amounts of wastewater with significant levels of chemical activation agents. Therefore, almost all the commercial processes for producing activated carbon use physical activation.

As mentioned above, the oil-containing sludge still has considerable value because of its residual hydrocarbons, and there are additional benefits in reducing the environmental risks of current waste management approaches such as incineration and sanitary landfill. On the other hand, there are few reports concerning the conversion of the oil-containing sludge into porous materials using the physical activation with gasification gas of carbon dioxide ($CO_2$) [14]. In this work, novel mesoporous Fe/carbon composites from oil-containing sludge were obtained by pyrolysis followed by $CO_2$ activation under the conditions of 850 °C and a range of residence times (30–90 min). We also treated the resulting carbon composites by dilute acid washing, which can leach inorganic remains from them for enhancing their pore properties. Subsequently, the pore properties of all carbon composites were determined by high-performance adsorption-desorption analyzer. In addition, the textures on the surface of carbon composites were observed by scanning electron microscopy (SEM).

## 2. Materials and Methods

### 2.1. Materials

The oil-containing sludge in the form of dark slurry was collected from the bottom of an oil-sludge separation unit in the local recycling plant (Pingtung County, Taiwan). The proximate analysis and calorific value of the as-received sludge sample was immediately determined to determine its properties with relevance to the potential for energy use. However, the analyses of organic and organic elements were performed on the dried oily sludge which had been dried in an oven for several days at approximately 70 °C.

### 2.2. Thermochemical Characteristics Analyses of Oil-Containing Sludge

#### 2.2.1. Thermogravimetric Analysis

In this work, thermogravimetric analysis (TGA) was performed to determine the thermal decomposition behavior of the sludge. As studied previously [15], the as-received sample (about 0.2 g) was placed into the TGA instrument (TGA-51; Shimadzu Co., Kyoto, Japan), which was heated at a ramp rate of 10 °C/min (ranging from room temperature to 1000 °C) under a nitrogen gas flow of 50 $cm^3$/min.

#### 2.2.2. Ash Content

The ash content of as-received oily sludge sample was determined using the American Society for Testing and Materials (ASTM) method ASTM D482 (i.e., "Standard Test Method for Ash from Petroleum Products") [16]. In this work, the measurement of moisture content was not determined because some organic matter remaining in the oily sludge could be released by thermal vaporization. To ensure the accuracy of the data, the determination of ash content was carried out in triplicate.

#### 2.2.3. Ultimate Analysis

For the use as a precursor for producing porous carbon composites, the contents of the oil-containing sludge in terms of its organic elements, including carbon (C), hydrogen (H), nitrogen (N), sulfur (S) and oxygen (O), were determined by an elemental analyzer (vario EL III; Elementar Co., Langenselbold, Germany). Using about 3 mg of the dried sludge sample, the determination of EA was performed in duplicate.

### 2.2.4. Heating Value Analysis

The heating value of the as-received oil-containing sludge sample was determined by a bomb-type calorimeter (CALORIMETER ASSY 6200; Parr Co., Moline, IL, USA), which was operated in isoperibolic mode (25 °C) in triplicate. Approximately 0.3 g of the sludge sample was used for each measurement.

### 2.2.5. Inorganic Element Analysis

The composition of inorganic elements in the dried oil-containing sludge samples were measured using an inductively coupled plasma-optical emission spectrometer (ICP-OES) (Agilent 725; Agilent Co., Santa Clara, CA, USA). In this work, ten relevant elements, including aluminum (Al), arsenic (As), barium (Ba), cadmium (Cd), chromium (Cr), copper (Cu), iron (Fe), lead (Pb), selenium (Se), and silicon (Si) were determined. Before performing the ICP-OES analysis, a concentrated acid solution (i.e., aqua regia solution) was used in a microwave-assisted digestion system to dissolve the dried sludge sample completely into a liquid matrix.

### 2.3. Pyrolysis-Activation Experiments

As reported previously [17,18], a vertical fixed-bed furnace was used to produce porous carbon composites using a one-pot process at a heating rate of about 10 °C/min. For each pyrolysis-activation experiment, about 3 g of dried oil-containing sludge was fed into the thermochemical system. The process conditions were set as follows: first, the pyrolysis was performed below 500 °C under a nitrogen gas flow rate of 500 $cm^3$/min. Then, the process was switched to the activation at elevated temperature (>500 °C) using a carbon dioxide gas flow rate of 100 $cm^3$/min. During the first stage, the vaporized gas was vented and mostly collected by a cooling system. The condensed oil product has a high heating value (*ca.* 42.0 MJ/kg). In this work, a series of experiments were performed by increasing the residence time from 30 to 90 min (30 min intervals) at an intermediate temperature of 850 °C because of the activation temperature in the range of 750–950 °C [14]. Furthermore, post-washing procedures with a dilute acid were performed to enhance the pore properties of the resulting carbon composites by removing inorganic matter from them. For this process, about 50 $cm^3$ of 0.25 M HCl solution and the crude carbon composite product were mixed together for 30 min on a hot-plate (set at about 75 °C). Following decantation, deionized water (100 $cm^3$) was added to rinse the residual slurry at the bottom of breaker, a process repeated three times to remove the residual inorganics. Prior to the pore analysis, drying of the post-washed carbon composite was carried out in an air-circulator. In order to describe the carbon composites simply, the resulting products were coded to allow simple comparisons of the data on the properties. For example, OS-CC-30 and OS-CC-30-A referred to the carbon composite (CC) products made from the oil-containing sludge (OS) at 850 °C for a holding time of 30 min without acid-washing and with acid-washing, respectively.

### 2.4. Analysis of Resulting Carbon Composites
### 2.4.1. Pore Analysis

In order to determine the pore development of the CC products, their nitrogen adsorption-desorption isotherms at −196 °C were measured in the ASAP 2020 Plus instrument (Micromeritics Co., Norcross, GA, USA). The pore properties (e.g., surface area and pore volume) can be calculated using the Brunauer–Emmett–Teller (BET) equation [19,20]. The device is a high-performance analyzer for measuring surface area, pore size, and pore volume of porous materials. Prior to the isotherm measurements, the sample was prepared to the degas stage by the removal of all impurities (i.e., moisture and light adsorbed matters) from the CC surface. This was accomplished in two stages: evacuation by vacuum pumping (≤1.33 Pa) and heating at elevated temperature (200 °C). Using the measured values of pressure, temperature and volume adsorbed, various models were used to calculate the pore properties of the porous material. The Langmuir and BET models

were used to indicate the single-point surface area, Langmuir surface area and BET surface area. The single-point surface area is a simplified case of the BET method, where the value of constant (i.e., *C*) is sufficiently large to ignore the value of intercept (i.e., *i*) in the BET equation. The t-plot method is a well-known technique which allows determination of the micro porous properties such as micropore (pore width < 2.0 nm) area and micropore volume. The Harrett–Joyner–Halenda (BJH) method was adopted to calculate the pore size distributions of the CC products in the mesopore (pore width: 2.0–50.0 nm) range. The total pore volume was estimated based on the assumption that all pores were filled by the condensed nitrogen (liquid density: 0.806 g/cm$^3$) at saturated relative pressure (i.e., 0.99). Therefore, it was obtained by converting the adsorbed nitrogen amount into liquid volume. Regarding the average pore width, it can be calculated from the data on the BET surface area and the total pore volume based on the cylindrical pore geometry.

### 2.4.2. Scanning Electron Microscopy

In order to validate the porous textures of the resulting CC products, their surface physical morphologies were observed by scanning electron microscopy (SEM) (S-3000N; Hitachi Co., Tokyo, Japan) under an accelerating voltage of 15.0 kV. Due to the non-conductance in the CC samples, they were first deposited with a thin gold film by an ion sputter (E1010; Hitachi Co., Tokyo, Japan) for providing good contrast in the SEM images.

## 3. Results and Discussion

### 3.1. Thermochemical Properties of Oil-Containing Sludge

Table 1 lists the data on the proximate analysis, elemental analysis and calorific value of the oily sludge used in the thermal activation experiments. It was found that the measured data were different from the results of other oil-containing or oily sludge feedstocks. For example, the contents of ash and carbon in the oil-containing sludge were about 13 wt% and 80.48 wt%, respectively, in the study by Mohammadi and Mirghaffari [7]. In other research by Yang et al. [21], the content of carbon was as high as 84.38 wt%. In this work, the sludge was clearly comprised of a large ash content (i.e., 56.01 wt%), causing a relatively lower calorific value (i.e., 18.01 MJ/kg). As expected, its carbon content (i.e., 34.34 wt%) was significantly lower than those in the published studies [7,21], in which the oily sludge was collected from the fraction at the bottom of oil storage tank in a petroleum refinery. The contents (i.e., 0.46 and 0.45 wt%, respectively) of nitrogen and sulfur in the oily sludge feedstock were relatively low in comparison with those (0.7–0.94 wt% and 0.7–1.41 for the contents of nitrogen and sulfur, respectively) of coal [22]. However, considerable emissions of nitrogen oxides (NOx), sulfur oxides (SOx) and particulates may be released into the atmosphere when treating it by direct combustion in waste incineration plants.

**Table 1.** Ash content, elemental analysis and calorific value of oil-containing sludge.

| Property | Value |
|---|---|
| Proximate analysis [a,b] | |
|     Ash (wt%) | 56.01 ± 0.78 |
| Ultimate analysis [b,c] | |
|     Carbon (wt%) | 34.34 ± 0.37 |
|     Hydrogen (wt%) | 4.46 ± 0.11 |
|     Oxygen (wt%) | 20.20 ± 0.07 |
|     Nitrogen (wt%) | 0.46 ± 0.10 |
|     Sulfur (wt%) | 0.45 ± 0.18 |
| Calorific value (MJ/kg) [a,b] | 18.10 ± 0.08 |

[a] The mean ± standard deviation for three determinations. [b] Air-dry basis (as received sample). [c] The mean ± standard deviation for two determinations.

Table 2 lists the contents of relevant inorganic elements, including Fe, Al, Cr and Cu. Clearly, the content of Fe was significantly higher than other inorganic elements in the oil-containing sludge sample, which was consistent with findings by Mohammadi and

Mirghaffari [7]. The high Fe content was likely connected with the rust contamination of the oil storage tank. Moreover, the concentrations of toxic elements (i.e., Ba, Cd, Pb, and Se) were not observed because they were lower than their corresponding method detection limits. However, the contents (i.e., 0.102 and 0.008 wt%, respectively) of Cr and Cu were found in the sludge, which may be contaminated by the alloy materials in the storage tank, pipeline and valve. The emissions of these toxic elements should be considered in the design and operation of thermochemical conversion systems.

**Table 2.** Contents of relevant trace elements in the oil-containing sludge.

| Inorganic Element | Value |
|---|---|
| Fe (wt%) | 20.077 |
| Al (wt%) | 0.390 |
| Cr (wt%) | 0.102 |
| Cu (wt%) | 0.008 |

Figure 1 shows the curves of the thermogravimetric analysis (TGA) and its derivative thermogravimetry (DTG) for the as-received oil-containing sludge sample at a heating rate of 10 °C/min under an inert environment with flowing nitrogen gas, which was normalized as compared to the initial sample mass. The thermal decomposition behavior of the sludge clearly showed three main regions. In the first stage, a weight loss of about 20% was observed in the range of 25–200 °C, which corresponds to the loss of moisture and thermal desorption of light volatile oils. The temperature interval for this stage was below 160 °C. The second stage involved a significant weight loss of about 35% in the temperature range of 200–500 °C. This change should be ascribed to the volatilization and pyrolytic decomposition of the heavy oil fractions in the oil-containing sludge. Therefore, the temperature at 500 °C was adopted to produce the porous carbon (char) in the first stage of the pyrolysis-activation experiments. The last stage showed a slight weight loss after 800 °C, which could be associated with the activation of the resulting char by the formed gases (e.g., $CO_2$) and the decomposition of inorganic compounds [23,24]. It is well known that the reaction between C (char) and $CO_2$ will be severe during the activation process [14,24]. Therefore, this work focused on the effects of residence time (i.e., 30–90 min) and acid-washing on the pore properties of the CC products produced at 850 °C.

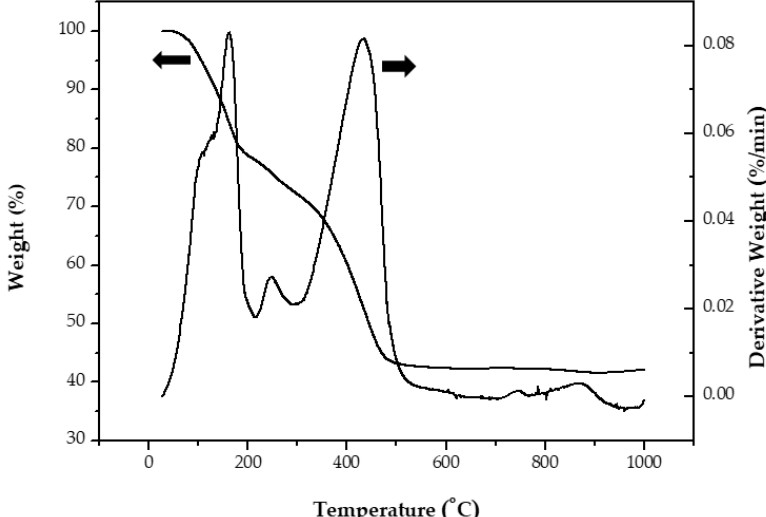

**Figure 1.** Thermogravimetric analysis (TGA) and derivative thermogravimetry (DTG) of oil-containing sludge at 10 °C/min under nitrogen flow.

### 3.2. Pore Analysis of Resulting Carbon Composites

The main pore properties (i.e., surface area, pore volume and pore size) of the resulting CC products are listed in Table 3. There was a consistent variation in the pore properties of the OS-CC products as a function of residence time from 30 to 90 min. The data indicated a clear decreasing trend with the increase in residence time at the activation temperature of 850 °C. For instance, the values of BET surface area showed a downward trend as residence time increased from 30 to 90 min as follows: 21.59 $m^2/g$ (OS-CC-30), 19.18 $m^2/g$ (OS-CC-60), and 0.56 $m^2/g$ (OS-CC-90). The products might be subject to continuous gasification due to the removal of limited carbon on the surface of activated char at longer residence time, thus causing the pore properties of the resulting CC products to be greatly reduced. On the other hand, the pore properties (using BET surface area for comparison) showed a consistent increase after acid-washing; that is, 40.53 $m^2/g$ (OS-CC-30-A), 28.98 $m^2/g$ (OS-CC-60-A), and 1.47 $m^2/g$ (OS-CC-90-A). The increase of specific surface area can be attributed to the dissolution of inorganic minerals by acid-washing. Therefore, the existing and new pores in the resulting CC products can be reformed and/or created, which has a significant effect on pore properties. In addition, the resulting CC showed an average adsorption efficiency of about 50% for removal of organics (total organic carbon $\approx$ 11,000 mg/L) from the industrial wastewater (i.e., the OS source).

**Table 3.** Pore properties of the resulting carbon composites.

| Pore Property | OS-CC-30 | OS-CC-30-A | OS-CC-60 | OS-CC-60-A | OS-CC-90 | OS-CC-90-A |
|---|---|---|---|---|---|---|
| Surface area ($m^2/g$) | | | | | | |
| Single point surface area [a] | 20.78 | 39.01 | 19.05 | 27.98 | 0.47 | 1.34 |
| BET surface area [b] | 21.59 | 40.53 | 19.18 | 28.98 | 0.56 | 1.47 |
| Langmuir surface area | 120.01 | 166.83 | 29.75 | 164.74 | 0.27 | 5.45 |
| *t*-plot micropore area [c] | 6.17 | 9.09 | 5.02 | 7.27 | 0.35 | 0.03 |
| *t*-plot external surface area | 15.42 | 31.44 | 14.15 | 21.71 | 0.21 | 1.44 |
| Pore volume ($cm^3/g$) | | | | | | |
| Total pore volume [d] | 0.050 | 0.072 | 0.054 | 0.067 | 0.002 | 0.005 |
| *t*-plot micropore area [c] | 0.003 | 0.004 | 0.003 | 0.003 | 0.0002 | 0.0000 |
| Pore size (nm) | | | | | | |
| Average pore width [e] | 9.32 | 7.14 | 11.22 | 9.22 | 13.94 | 14.50 |

[a] Calculated at a relative pressure of 0.30 using the BET method. [b] Calculated in a relative pressure range of 0.06–0.30 (9 points) using the BET method. [c] Using the *t*-plot method. [d] Calculated at a relative pressure of 0.995 (i.e., saturated adsorption). [e] Obtained by the ratio of the total pore volume ($V_t$) to the BET surface area ($S_{BET}$) assuming the cylindrical pore geometry (i.e., average pore width = $4 \times V_t/S_{BET}$).

Figure 2 shows the $N_2$ adsorption-desorption isotherms of the optimal CC (i.e., OS-CC-30) and its acid-washed CC (i.e., OS-CC-30-A) as a function of relative pressure ($P/P_0$), which is the equilibrium pressure (P) divided by the saturation pressure ($P_0$). According to the classification by the International Union of Pure and Applied Chemistry (IUPAC) [22], the type IV isotherms are typically characteristic of mesoporous materials because of the existence of the hysteresis loop. This loop at a relative pressure of 0.4–1.0 was observed, implying that the presence of mesoporous structure in the CC products. Furthermore, the type H3 hysteresis loop as described in the IUPAC classification was seen in the resulting CC products, indicating the existence of slip-shaped pores in the CC products. The desorption branch for the type H3 hysteresis loop also contained a steep region, which occurred at −196 °C in the relative pressure range from 1.0 to 0.4. Making a comparison between the data in Table 3 and the positions in Figure 2, it was consistent because of the isotherm curves of the acid-washed CC (i.e., OS-CC-30-A) positioned on the upper side. In this work, the optimal CC products (i.e., OS-CC-30 and OS-CC-30-A) are mesoporous materials based their pore size distributions, which were observed in the pore size range of 3.0 to 4.5 nm (Figure 3).

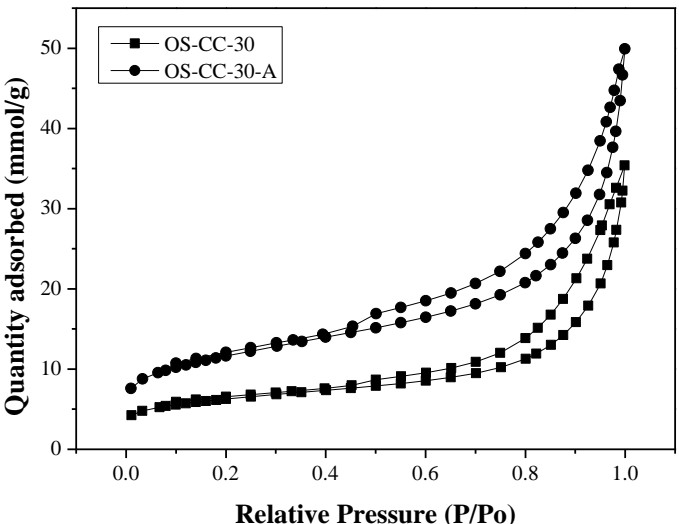

**Figure 2.** Nitrogen adsorption-desorption isotherms for the CC products (i.e., OS-CC-30 and OS-CC-30-A).

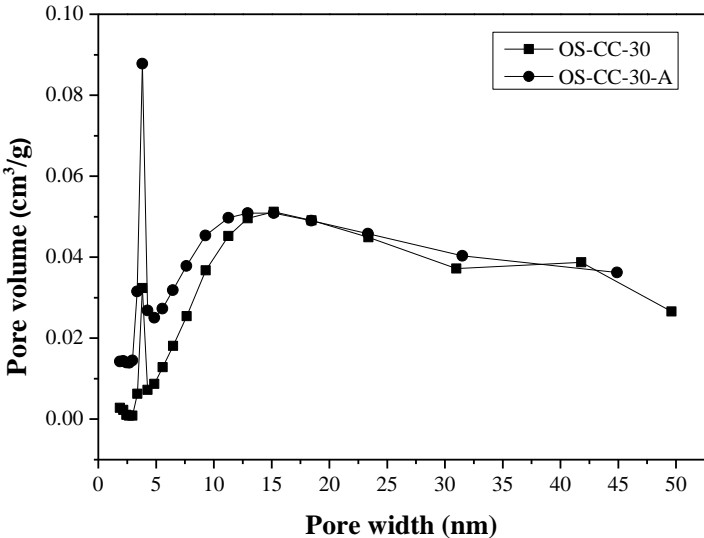

**Figure 3.** Pore size distributions for the CC products (i.e., OS-CC-30 and OS-CC-30-A).

Figure 4 shows the SEM images of the optimal CC products (i.e., OS-CC-30 and OS-CC-30-A). Pitted and heterogeneous structures were clearly evident. The image in Figure 4a, shows many grains on the surface, which could be derived from the inorganic minerals such as iron oxides. By contrast, the post-washing generated more pores on the clean surface of the acid-washed CC product (i.e., OS-CC-30-A) in Figure 4b. The SEM observations were also consistent with their pore properties in Table 2. Further, the resulting CC products also feature magnetic and hydrophilic properties due to the presence of oxygen-containing functional groups and iron oxides on the surface. In this regard, the resulting mesoporous Fe/CC products could be an effective adsorbent for removing pollutants from aqueous solution and also easily separated and recycled by external magnets [25].

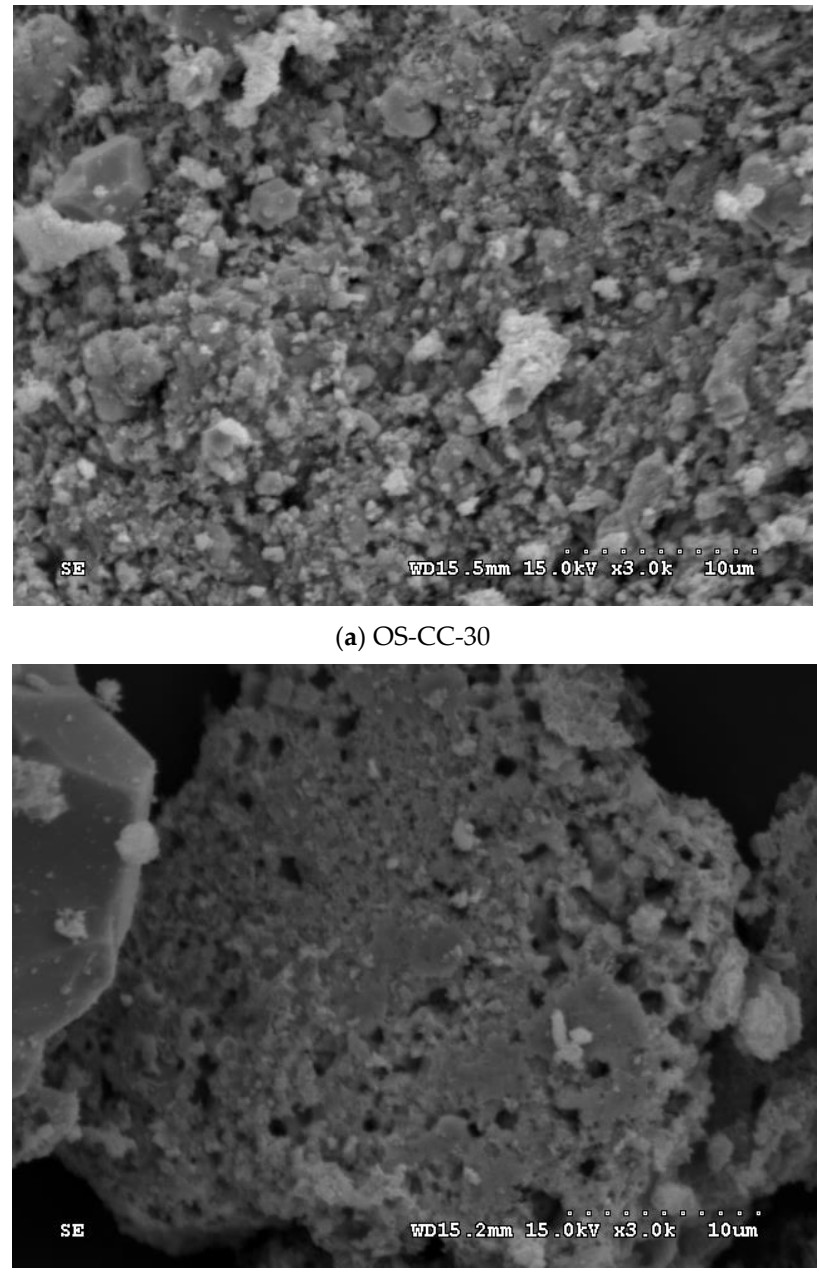

(**a**) OS-CC-30

(**b**) OS-CC-30-A

**Figure 4.** Images of SEM ($\times$3000) for the CC products of (**a**) OS-CC-30 and (**b**) OS-CC-30-A.

## 4. Conclusions

In the present work, the reuse of oil-containing sludge (OS) as a potential feedstock for the production of mesoporous magnetic carbon composite (CC) has been studied in a pyrolysis-activation process. The preparation experiments were set at 850 $^{\circ}$C as a function of residence time (30–90 min) based on the thermal decomposition behavior of the OS feedstock by the thermogravimetric analysis (TGA). The findings indicate that the pore properties greatly decreased from 21.59 m$^2$/g at the residence time of 30 min to 0.56 m$^2$/g at the residence time of 90 min. Such a decline may be due to continuous gasification by $CO_2$ at the longer residence time, thus enhancing the removal of limited carbon from the surface of activated char in the form of gas products (e.g., CO). In addition, the post-treatment of the CC products by acid-washing was further performed to remove part of the ashes and/or inorganic minerals. Thus, the positive effect on the pore properties significantly promoted the development of the porous structure in the acid-washed CC

products, showing an increase of BET surface area from 21.59 to 40.53 m$^2$/g at the residence time of 30 min. More obviously, the resulting CC products from the pyrolysis-activation process and acid-washing post-treatment are mesoporous materials, which could be a good adsorbent for removal of adsorbates with high molecular weight (or large molecular size) from an aqueous environment. Due to the richness in iron oxides, the exhausted CC products could be easily separated and further recycled by applying external magnets. This $CO_2$-activation process may be extended to discover a novel approach for the production of porous materials made from a variety of organic sludge residues.

**Author Contributions:** Conceptualization, W.-T.T.; methodology, Y.-Q.L.; validation, Y.-Q.L.; data curation, Y.-Q.L.; formal analysis, Y.-Q.L.; resources, W.-T.T.; writing—original draft preparation, W.-T.T.; writing—review and editing, W.-T.T.; supervision, W.-T.T. All authors have read and agreed to the published version of the manuscript.

**Funding:** This research was funded by the Ministry of Science and Technology (Taiwan), grant number MOST 109-2622-E-020-004.

**Institutional Review Board Statement:** Not applicable for studies not involving humans or animals.

**Informed Consent Statement:** Not applicable for studies not involving humans.

**Data Availability Statement:** Not applicable.

**Acknowledgments:** The authors express sincere appreciation to the Instrument Center of National Pingtung University of Science and Technology for the assistance in the scanning electron microscope (SEM) analysis. We also appreciate the assistance in elemental analysis (EA) provided by the Instrument Center of National Chung Hsing University.

**Conflicts of Interest:** The authors declare no conflict of interest.

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
