# Peer review of "Preparation and Characterization of Porous Carbon Composites from Oil-Containing Sludge by a Pyrolysis-Activation Process"

_processes, doi:10.3390/pr10050834_

Round 1
Reviewer 1 Report
The present manuscript is prepared porous carbon composites from oil containing sludge and characterized them. Considering the scientific soundness and novelty, the manuscript failed to attain the quality to claim the publication with the present version. However the impact of the manuscript can be improved with the following comments.
- The prepared composite should be presented with any of the potential applications. As the authors have mentioned the application of adsorbent, preliminary investigation on the adsorption potential should be included to provide the significance of the prepared material.
2. Also the surface topography should be analyzed with appropriate techniques besides SEM analysis.
Author Response
Q1. The prepared composite should be presented with any of the potential applications. As the authors have mentioned the application of adsorbent, preliminary investigation on the adsorption potential should be included to provide the significance of the prepared material.
Reply: We agree to the comment on the investigation on the adsorption potential. A preliminary result for the adsorption potential has been added to the first paragraph in Sec. 3.2 as follows.
“In addition, the resulting CC preliminarily showed the average adsorption efficiency of about 50% for removal of organics (total organic carbon ≈ 11,000 mg/L) from the industrial wastewater (i.e., OS source).”
Q2. Also the surface topography should be analyzed with appropriate techniques besides SEM analysis.
Reply: Surface topography generally refers to the profile shape and the surface roughness of a sample, which play a vital role in the capacities of load-carrying, wear and lubrication. However, the SEM analysis in this work was performed to observe the porous textures, which were used to verify the pore properties of the resulting carbon composites (Table 3) from the nitrogen adsorption-desorption measurements.
Reviewer 2 Report
The manuscript provides an environmentally friendly and applicable way to handle oil-containing sludge. There are several points ay needed to clarify before publishing.
Firstly, the oil sludge was collected in a recycling plant, do the properties vary with collecting in a different season or even different month? As the aim of this manuscript is to produce mesoporous magnetic carbon composite, what is the meaning of determining calorific value? Contents of Fe may affect the magnetic properties of the products. It is also stated in the results part that Fe/CC products can be separated by external magnets. Are there any experiences or literature that support the statement? Moreover, does the variation of Fe content affect the separating properties? Lastly, the pore volume of lower pore width has a sharp increase in acid wash treated oil sludge, what are the reasons behind?
Author Response
Q1. The oil sludge was collected in a recycling plant, do the properties vary with collecting in a different season or even different month?
Reply: Indeed, the properties of oil-containing sludge from the waste oil recycling plant varies greatly, depending on its collection time and objects (generation sources) such as petroleum-refining, metal-working or vehicle-repairing. In this work, the oil-containing sludge should be derived from the tank bottoms in the petroleum refinery due to its high Fe content (i.e., 21.1 wt%, as seen in Table 2).
Q2. As the aim of this manuscript is to produce mesoporous magnetic carbon composite, what is the meaning of determining calorific value?
Reply: The pore properties of resulting mesoporous magnetic carbon composites are highly related to the carbon level of the oil-containing sludge. Furthermore, the calorific value is a measure of the energy content released upon the complete combustion of the oil-containing sludge under oxygen atmosphere. In general, there is a positive relationship between the carbon content and the calorific value of a sample.
Q3. Contents of Fe may affect the magnetic properties of the products. It is also stated in the results part that Fe/CC products can be separated by external magnets. Are there any experiences or literature that supports the statement? Moreover, does the variation of Fe content affect the separating properties?
Reply: We referred to the easy separation of Fe/CC products by external magnets in the Ref. 43. We also experienced this feature from the aqueous solution using a magnetic rod made of PTFE. Regarding the effect of Fe content in the resulting CC on the separating properties, we did not perform the test.
Q4. The pore volume of lower pore width has a sharp increase in acid wash treated oil sludge, what are the reasons behind?
Reply: As shown in Fig. 3, the pore volume of lower pore width (Ë‚ 2 nm) has a sharp increase, implying the existence of micropores in a small portion.
Reviewer 3 Report
Good day! My review is in the attached file.

Author Response
Q1. There are a few comments about the text: in Figure 2, there is no explanation of what P and P0 are.
Reply: The terms of P and P0 in Figure 2 have been explained as follows.
“Figure 2 further depicted the N2 adsorption-desorption isotherms of the optimal CC (i.e., OS-CC-30) and its acid-washed CC (i.e., OS-CC-30-A) as a function of relative pressure (P/P0), which is the equilibrium pressure (P) divided by the saturation pressure (P0).”
Q2. The next question deals with experimental conditions. I think there is a need to justify the accepted time interval for pyrolysis of used oils of 30 to 90 minutes. Perhaps 20 to 25 minutes would be even better?
Reply: We thank the Reviewer’s comment on the residence time conditions in the pyrolysis-activation experiments. In this work, a series of experiments were performed by increasing the residence time from 30 to 90 min (30 min intervals) at an intermediate temperature of 850°C because of the commonly used activation in the temperature range of 750-950°C. In order to produce the mesoporous magnetic carbon composites with higher pore properties, it would be helpful to perform the experiments at lower residence time (i.e., 0-30 min).
Q3. The final composition and quantity of wash solutions after the scrubbing of sorbent samples are not specified here. In addition, there is no show the change in the pH of the material after treatment with an acidic solution.
Reply: In this work, the crude carbon composite product was further treated using about 50 cm3 of 0.25 M HCl solution for mixing on a hot-plate (about 75°C for 30 min). After decanting the upper solution, the bottom slurry was rinsed with deionized water (100 cm3) for three times to remove the residual inorganics. Based on the previous experience, the pH value in the final decanted solution was close to 7.0. Regarding the compositions of wash solutions after the scrubbing of the resulting CC samples, they should contain the elements of Fe and Al based on the energy dispersive X-ray spectroscopy (not shown here) and the data in Table 2.
Q4. The manuscript does not present experiments on the extraction of contaminants from aqueous solutions with the sorbent obtained, or at least references to sources of this information.
Reply: In this work, the crude carbon composite product was further treated using about 50 cm3 of 0.25 M HCl solution for mixing on a hot-plate (about 75°C for 30 min). After decanting the upper solution, the bottom slurry was rinsed with deionized water (100 cm3) for three times to remove the residual inorganics. Regarding the experiment on the extraction of contaminants from aqueous solutions, it did not be performed because the aqueous solution should be non-hazardous and also has no economic value based on its contained ions (i.e., Fe and Al).
Q5. The question of the release of secondary contaminants (sludge pyrolysis products) to water is not discussed here.
Reply: In the first stage of the pyrolysis-activation experiments, the pyrolysis was performed below 500°C under the nitrogen gas flow rate of 500 cm3/min. The sludge pyrolysis products mainly included the vaporized/vented gas, which was condensed by a cooling system for the production of fuel oil or pyrolytic oil with a high calorific value (ca. 42.0 MJ/kg). The above-mentioned description has been added as follows.
“During the first stage, the vaporized gas was vented and mostly collected by a cooling system. The condensed oil product has a high heating value (ca. 42.0 MJ/kg).”
Q6. There is no preliminary economic justification for the feasibility of the disposal of the waste under consideration and the production of these sorbents in this way.
Reply: In the present study, the main purpose was to develop an environmentally friendly and applicable way to handle oil-containing sludge. In this regard, the economic evaluation was not the focus, but it should be addressed in the further production plan.

Reviewer 4 Report
The paper concerns very impotrant problem of utilization of oil-containg sludge to produce porous carbo composite. The work presents a typical experimental tests on the material from Taiwan recycling plants. I have some remarks:
- Are the presented test results applicable to all types of oil containing sludge in the world?
- How can the presented test methods be applied to industrial applications? A few tasks on this topic would be nice.
- It would be nice shortly describe BET calculation model because the references relates to books and potential readers will want know about such method.
- It is good practice to put the used abbreviations with their explanation at the end of the article, before the reference list.
- line 121 - what means M?
- line 256 - should be "Figure 4 shows..."
This short remarks are intended to improve the paper and interest the readers. After small corrections the paper should be publish in the Journal.
Author Response
Q1. Are the presented test results applicable to all types of oil containing sludge in the world?
Reply: Indeed, the properties of oil-containing sludge from the waste oil recycling plant varies greatly, depending on its collection time and objects (generation sources) such as petroleum-refining, metal-working or vehicle-repairing. In this work, the oil-containing sludge should be derived from the tank bottoms in the petroleum refinery due to its high Fe content (i.e., 21.1 wt%, as seen in Table 2).
Q2. How can the presented test methods be applied to industrial applications? A few tasks on this topic would be nice.
Reply: We thank the Reviewer’s comment on the industrial applications. In this work, the pyrolysis-activation process is a commonly used method for the production of porous materials (i.e., activated carbon) from the carbon-containing precursors like lignocelluloses, hydrocarbons and polymers or fibers.
Q3. It would be nice shortly describe BET calculation model because the references relates to books and potential readers will want know about such method.
Reply: We thank the reviewer for the positive comment. The description about the BET method has been added as follows.
“In order to determine the pore development of the CC products, their nitrogen adsorption-desorption isotherms at – 196°C were measured in the ASAP 2020 Plus instrument (Micromeritics Co., USA). The pore properties (e.g., surface area and pore volume) can be calculated using the Brunauer-Emmett-Teller (BET) equation [39, 40].”
Q4. It is good practice to put the used abbreviations with their explanation at the end of the article, before the reference list.
Reply: According to the Instructions for Authors, the used abbreviations should be defined the first time they appear in each of three sections: the abstract; the main text; the first figure or table. Therefore, the manuscript followed this style without their explanations at the end of the article.
Q5. line 121 - what means M?
Reply: It is the molar concentration symbol (mol/L) in SI unit.
Q6. line 256 - should be "Figure 4 shows..."
Reply: It has been corrected in the revised manuscript.
Round 2
Reviewer 1 Report
The authors have revised considerably
Author Response
Q1. The authors have revised considerably.
Reply: We thank the reviewer for the positive comment.